# Report

EMBO
Molecular Medicine

# Gut microbiota influences pathological angiogenesis in obesity-driven choroidal neovascularization

Elisabeth MMA Andriessen[1], Ariel M Wilson[2], Gaelle Mawambo[3], Agnieszka Dejda[3,4], Khalil Miloudi[5], Florian Sennlaub[6] & Przemyslaw Sapieha[1,3,4,5,*]

## Abstract

**Age-related macular degeneration in its neovascular form (NV AMD) is the leading cause of vision loss among adults above the age of 60. Epidemiological data suggest that in men, overall abdominal obesity is the second most important environmental risk factor after smoking for progression to late-stage NV AMD. To date, the mechanisms that underscore this observation remain ill-defined. Given the impact of high-fat diets on gut microbiota, we investigated whether commensal microbes influence the evolution of AMD. Using mouse models of NV AMD, microbiotal transplants, and other paradigms that modify the gut microbiome, we uncoupled weight gain from confounding factors and demonstrate that high-fat diets exacerbate choroidal neovascularization (CNV) by altering gut microbiota. Gut dysbiosis leads to heightened intestinal permeability and chronic low-grade inflammation characteristic of inflammaging with elevated production of IL-6, IL-1β, TNF-α, and VEGF-A that ultimately aggravate pathological angiogenesis.**

**Keywords** age-related macular degeneration; angiogenesis; gut microbiota; inflammaging; obesity

**Subject Categories** Metabolism; Immunology; Neuroscience

See also: **R Scholz & T Langmann** (December 2016)

## Introduction

While AMD is the leading cause of irreversible blindness in the industrialized world (Klein & Klein, 2004; Maberley *et al*, 2006; Wong *et al*, 2014), the cellular and molecular mechanisms that precipitate disease remain incompletely understood despite significant genomewide association studies identifying susceptibility genotypes and target mechanistic pathways (Fritsche *et al*, 2013, 2016; Sobrin & Seddon, 2014; Black & Clark, 2016). Obesity has long been suspected as a risk factor for AMD, but increased body mass index associations with AMD were inconsistent (Zhang *et al*, 2016). Using 21,287 participants from the Melbourne Collaborative Cohort Study, it was recently demonstrated that each increase of 0.1 in waist/hip ratio (a measure for abdominal obesity) was associated with a 13% increase in the odds of early AMD and a 75% increase in the odds of late AMD in men, making obesity the second most important environmental risk factor for late AMD after cigarette smoking (Adams *et al*, 2011).

Progression of AMD is influenced by single or compounded environmental and genetic risk factors that lead to persistent low-grade inflammation and a largely innate immune response (Combadière *et al*, 2007; Ambati *et al*, 2013; Sennlaub *et al*, 2013). Evidence for environmental factors predisposing to AMD is supported by the fact that genetically unrelated individuals with shared long-term environmental exposure develop the disease with a concordance of 70.2% (Gottfredsdottir *et al*, 1999). A consequence of cohabitation and common lifestyle habits that prospectively impact disease modifiers such as systemic inflammation is microbial exchange (Song *et al*, 2013). Given that commensal gut microbiota exert profound influence on digestion, dietary metabolism, endotoxemia, and immune responses (Backhed *et al*, 2005; Turnbaugh *et al*, 2006; Cani *et al*, 2008; Cerf-Bensussan & Gaboriau-Routhiau, 2010), they are prime candidates to impact chronic low-grade inflammation (Tremaroli & Backhed, 2012). Microbiota-related low-grade inflammation is characterized by elevated pro-inflammatory gene expression and is a common consequence of an altered host–microbiota relationship caused by instigator bacteria or dietary components that influence intestinal permeability (Chassaing & Gewirtz, 2014). There is accumulating evidence that asserts the importance of intestinal permeability, a barrier aspect closely associated with the intestinal commensal microbiota as well as with the mucosal immune system, in intestinal and systemic health (Brun *et al*, 2007;

1   Department of Biomedical Sciences, Maisonneuve-Rosemont Hospital Research Centre, University of Montreal, Montreal, Quebec, Canada
2   Department of Engineering Physics, Laser Processing and Plasmonics Laboratory, École Polytechnique de Montréal, Montreal, Quebec, Canada
3   Department of Biochemistry, Maisonneuve-Rosemont Hospital Research Centre, University of Montreal, Montreal, Quebec, Canada
4   Department of Ophthalmology, Maisonneuve-Rosemont Hospital Research Centre, University of Montreal, Montreal, Quebec, Canada
5   Department of Neurology-Neurosurgery, McGill University, Montreal, Quebec, Canada
6   INSERM, U 968, Sorbonne Universités, Université Pierre et Marie Curie Paris 06, Unité Mixte de Recherche S 968, Institut de la Vision, CNRS, Unité Mixte de Recherche, Paris, France
    *Corresponding author. Tel: +1 514 252 3400 x7711; Fax: +1 514 252 3569; E-mail: mike.sapieha@umontreal.ca

Manco et al, 2010; Gerova et al, 2011; Neves et al, 2013; Bischoff et al, 2014). Heightened intestinal permeability can allow for an increased translocation of bacterial products such as LPS and other pathogen-associated molecular pattern molecules (PAMPs) (Cani et al, 2008; Cerf-Bensussan & Gaboriau-Routhiau, 2010; Mehal, 2013). These PAMPs impact pro-inflammatory signaling through pattern recognition receptors (PRRs) of the innate immune system, especially the Toll-like receptors (TLRs) and Nod-like receptors (NLRs), inducing low-grade systemic inflammation (Chassaing & Gewirtz, 2014). Microglia, perivascular macrophages, a small number of dendritic cells, and RPE cells express various PRRs, including TLRs and NLRs, and may predispose for PAMPs of intestinal origin to impact ocular inflammation (Chen & Xu, 2015). In addition, bacterial products or metabolites from gut microbiota can modulate microglia maturation, morphology, and function (Erny et al, 2015) and activate retina-specific T cells that are thought to be involved in autoimmune uveitis (Horai et al, 2015).

Dysbiosis in gut microbes is particularly important for the aging population given that they modulate aging-related changes in innate immunity, sarcopenia, cognitive function, and frailty in general (O'Toole & Jeffery, 2015). Here, we sought to evaluate the contribution of intestinal flora to progression of NV AMD, particularly in the context of obesity-driven CNV.

## Results

### High-fat diet modulates gut microbiota and exacerbates CNV

In light of epidemiologic data linking obesity to CNV (Seddon et al, 2003; Zhang et al, 2016), we first investigated the propensity of diets with elevated fat content to exacerbate CNV. C57BL/6J mice were raised under sterile barrier conditions and placed on a regular-chow diet (RD; 16% kcal fat) or high-fat diet (HFD; 60% kcal fat), from 6 weeks of age (Fig 1A). As expected, upon killing at 13 weeks, HFD-fed mice gained over 50% more weight than control RD-fed mice (Fig 1B and C). At 11 weeks of life, we subjected mice to a laser-induced photocoagulation model of CNV, where perforation of Bruch's membrane initiates sprouting of subretinal blood vessels from the choroid, thus mimicking NV AMD (Lambert et al, 2013). Quantification of FITC–dextran-perfused neovessels over isolectin B4 (IB4)-labeled impact area by confocal imaging 14 days

after laser burn revealed a robust 60% increase in CNV in HFD-fed mice when compared to RD controls (Fig 1D and E). These experimental mouse data are in line with previously published results (Sene et al, 2013) and verify human studies (Seddon et al, 2003; Adams et al, 2011).

To tease out the contribution of gut microbes to heightened CNV in mice receiving HFD, mice were administered 0.5 g/l of the broad-spectrum, non-gut permeable (Cani et al, 2008) antibiotic (AB) neomycin trisulfate salt hydrate orally through their drinking water. Remarkably, HFD-fed mice treated with neomycin displayed levels of CNV akin to RD-fed control mice and thus significantly less than HFD-fed mice receiving vehicle control (Fig 1D and E). Treatment with neomycin did not impact weight gain (Fig 1B and C), hence uncoupling mouse weight from extent of CNV and strengthening the link between gut flora and pathological angiogenesis.

To ascertain that HFD and oral neomycin modify intestinal flora, we next profiled gut microbiomes. The ratio of Bacteroidetes and Firmicutes, the two dominant phyla that make up over 90% of bacterial phylogenetic types in the distal gut (Ley et al, 2006), is affected by diet (Turnbaugh et al, 2009; David et al, 2014). Comparison of distal gut microbiota from lean and obese individuals shows that the relative proportion of Bacteroidetes is decreased in obese individuals compared to lean individuals (Turnbaugh et al, 2006; David et al, 2014), which is consistent with mouse studies and underscores that diet impacts intestinal flora (Ley et al, 2006; Turnbaugh et al, 2006; Hildebrandt et al, 2009). We characterized gut microbiome composition by sequencing the hypervariable regions V2, V3, V4, V6, V7, V8, and V9 of bacterial 16S rRNA extracted from fecal pellets originating from RD-fed and HFD-fed mice receiving vehicle, as well as HFD-fed and RD-fed mice receiving neomycin (Figs 1F and EV1). Consistent with clinical data (Ley et al, 2006; Turnbaugh et al, 2006; Wu et al, 2013), mice on HFD had shifted ratios of commensal gut microbes. Bacteroidetes/Firmicutes ratios shifted from 66%/33% of total bacteria in RD to 19%/67% in HFD. Importantly, oral neomycin restored the proportion of Bacteroidetes to ~65% of total bacteria and reduced Firmicutes from ~65% to < 10%. In addition, relative abundance of Proteobacteria (a microbial signature of dysbiosis in gut microbiota (Wu et al, 2013)) rose with HFD and further with antibiotic treatment reaching 25% of the population (Fig 1G). Of note, HFD-fed mice host the most diverse microbiome, with a modest but important presence of Actinobacteria and Spirochaetes (0.5 and 1.5% of the total). These phyla were

▶

---

**Figure 1.  High-fat diet exacerbates CNV and influences gut microbiota.**

A     Schematic representation of experimental timeline where half of the mice start a high-fat diet (HFD) at 6 weeks and later half of these receive neomycin (AB) treatment from the age of 9 weeks until killing at week 13. Control mice were fed a regular-chow diet (RD). At the age of 11 weeks, mice are subjected to four laser burns per eye to perforate Bruch's membrane and recruit subretinal blood vessels from the choroid.

B, C   Weight gain (B) and area under the curve (C) of percentage weight gain of HFD-fed mice compared to RD-fed mice, treated with vehicle or antibiotic; n = 13 (RD), 17 (HFD), 16 (HFD+AB), 12 (RD+AB); CI 99.9%.

D     Compilation of compressed Z-stack confocal image of FITC–dextran-labeled CNV and single-plane confocal image of isolectin B4-stained choroidal flat mounts from RD- and HFD-fed mice, with vehicle or antibiotic treatment.

E     Quantification of area of FITC–dextran-labeled CNV over isolectin B4-stained laser impact area fold RD; n = 11 (RD), 16 (HFD), 11 (HFD+AB), 12 (RD+AB), with 36 (RD), 32 (HFD), 33 (HFD+AB), 34 (RD+AB) burns total; CI 99%.

F, G   Representative circle charts of relative abundance of bacterial phyla, class, order, and family (from central to peripheral), in gut microbiota of RD-fed mice with vehicle, HFD-fed mice with vehicle, HFD-fed mice with neomycin, and RD-fed mice with neomycin (F) and relative proportion per group of different phyla; n = 7 (RD), 8 (HFD), 6 (HFD+AB), 7 (RD+AB); Bacteroidetes; CI 99.9%, Firmicutes; CI 99%, Proteobacteria; CI 99%, Actinobacteria and Spirochaetes (G).  *n.d. = not detected.

Data information: All comparisons between groups are analyzed using one-way analysis of variance (ANOVA) and Tukey's multiple comparisons test; **P < 0.01, ***P < 0.001; error bars represent mean ± SEM. Each "n" represents one mouse per experimental group; CI, confidence interval. Scale bar: 100 μm.

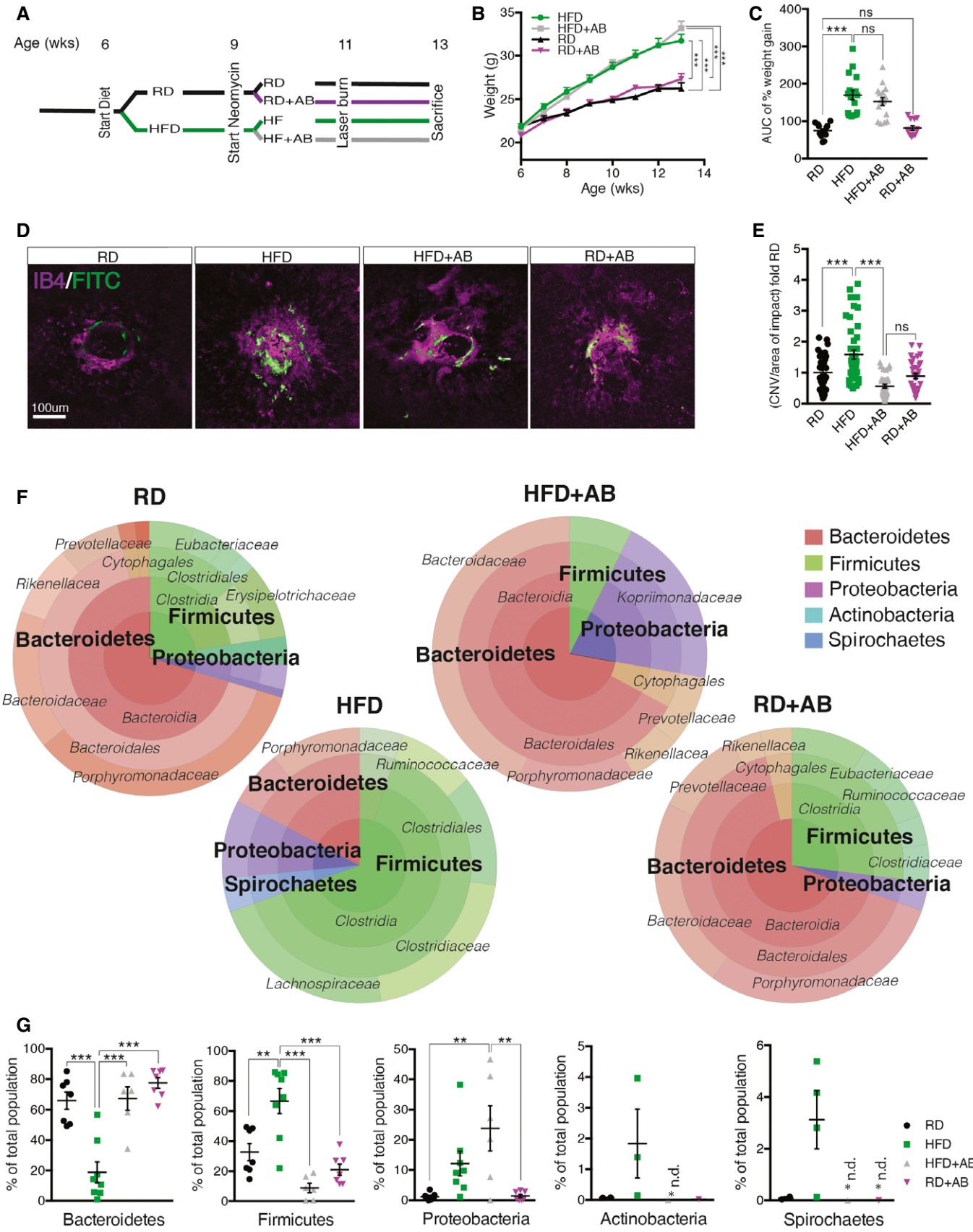

Figure 1.

undetectable in RD-fed mice or after receiving oral neomycin. RD mice on neomycin showed consistently higher levels of Proteobacteria at the expense of Firmicutes, confirming microbial shifts with oral neomycin. Together, these data reconcile that modulation in gut microbiota correlates with severity of CNV.

### High-fat diet potentiates recruitment of microglia and macrophages during CNV

Mononuclear phagocytes (MPs) such as microglia and macrophages contribute to AMD pathogenesis (Ambati et al, 2003b; Combadiere et al, 2007; Langmann, 2007) and localize to sites of neovascularization following laser-induced CNV (Combadière et al, 2007; Liu et al, 2013; Indaram et al, 2015; Luckoff et al, 2016) (Fig 2A–C). Microglia are CNS-resident macrophages that are derived from primitive macrophages in the yolk sac (Prinz et al, 2014). Upon CNS injury, they move to sites of insult and polarize to an activated state and produce pro-inflammatory cytokines (Hanisch & Kettenmann, 2007). Depending on the extent of injury, circulating monocytes may also be recruited from the bloodstream.

Analysis by FACS of whole retinas and RPE–choroid–sclera complexes from mice on regular diets revealed a twofold rise in MPs (Ly6G$^-$, F4/80$^+$, CD11b$^+$) at 3 (p3) and 7 (p7) days post-laser burn (Fig 2D and E) when compared to non-burned eyes (gating scheme in Fig EV2A). At day 14 post-burn (p14), the number of MPs returned to basal levels (Fig 2D and E). A similar kinetic profile was observed for microglia (Ly6G$^-$, F4/80$^+$, CD11b$^+$, CX3CR1$^{hi}$ CD45$^{lo}$) (Fig 2F and G). Notably, the overall proportion of microglia within the MP population dropped in the first week following laser burn (p3 and p7), suggesting cellular infiltration from circulation (Fig EV2B).

Importantly, a HFD potentiated these effects as the heightened CNV observed in HFD-fed mice (Fig 1D and E) was accompanied by a ~twofold increase in MPs and microglia at sites of lesion when compared to control RD-fed mice (Fig 2H–K). This increase in subretinal MPs persisted at 14 days post-burn as determined by immunofluorescent quantification of IBA-1$^+$ cells around the laser-induced lesion (Fig 2L and M). Oral neomycin abrogated this recruitment (Fig 2H–M), which is consistent with the observed reduction in CNV (Fig 1D and E). Hence, the intestinal microbiota of HFD-fed mice augments recruitment of microglia and other MPs that may be central to disease progression.

### Heightened CNV in mice with dysbiosis is accompanied by increased intestinal permeability, metabolic endotoxemia, and systemic inflammation

Pattern recognition receptors (PRRs) allow the innate immune system to recognize pathogen-associated molecular patterns (PAMPs) and trigger an inflammatory reaction to fight off the microbes that produce them (Dorrestein et al, 2014). Although PAMPs circulate at low concentrations under physiologic conditions (DiBaise et al, 2012), an increase in LPS concentrations, a condition termed "metabolic endotoxemia", may provoke low-grade inflammation (Cani et al, 2008), insulin resistance (Cani et al, 2007), augmented cardiovascular risk (Manco et al, 2010), fatty liver disease (Brun et al, 2007), white adipose tissue inflammation (Caesar et al, 2015), and retinopathy of prematurity (ROP) (Tremblay et al, 2013). In obesity, changes in gut microbiota have been suggested to compromise barrier function of the epithelial layer of the gut, thus increasing entry of PAMPs into systemic circulation (Amar et al, 2008; Cani et al, 2008; Osborn & Olefsky, 2012).

Analysis of intestinal permeability by Evans Blue assay revealed that HFD elevated intestinal permeability by threefold when compared to controls (Fig 3A). In our experimental paradigm, antibiotic treatment did not restore intestinal permeability of HFD-fed mice as there was no significant difference between HFD-fed mice and HFD-fed mice that were treated with antibiotics (Fig 3A).

We next determined whether the levels of circulating PAMPS were sufficient to trigger a PRR response. We therefore subjected murine macrophages that express several PRRs with a chromosomal integration of a secreted embryonic alkaline phosphatase reporter construct inducible by NF-κB and AP-1 (RAW-Blue™ cells) to serum from mice under various feeding paradigms. This reports on activation of PRRs such as Toll-like receptors (TLRs), NOD-like receptors (NLRs), RIG-I-like receptors (RLRs), and C-type lectin receptors (CLRs).

Serum from HFD-fed mice activated PRRs significantly more readily than serum from RD-fed mice as had been suggested (Cani et al, 2008; Membrez et al, 2008; Neves et al, 2013) or HFD-fed mice receiving antibiotic treatment (Fig 3B). The reduction in PRR response in antibiotic-treated mice may occur due to an overall reduction in the absolute number of bacteria present in the intestine.

We next investigated both systemic and local profiles of classic inflammatory mediators that are associated with para-inflammation in AMD (Chen & Xu, 2015). Consistent with the heightened

---

**Figure 2.  High-fat diet increases recruitment of microglia and macrophages.**

A–C   3D rendering of Z-stack confocal image showing isolectin B4-stained laser burn with FITC–dextran-labeled CNV and IBA-1-stained MPs.

D   Representative FACS plots of retinas and sclera–choroid–RPE cell complexes from regular-chow diet (RD)-fed mice without and 3, 7, and 14 days after laser burn.

E   Quantification of MPs (Ly6G$^-$, F4/80$^+$, CD11b$^+$) at p3, p7, and p14; fold naïve (no burn); n = 5 (no burn), 5 (p3), 6 (p7), 5 (p14); CI 95%.

F, G   Representative FACS plots (F) and quantification of microglia (G) (Ly6G$^-$, F4/80$^+$, CD11b$^+$, CX3CR1$^{hi}$ CD45$^{lo}$) fold naïve; n = 5 (no burn), 4 (p3), 6 (p7), 5 (p14); CI 95%.

H   Representative FACS plots of retinas and sclera–choroid–RPE cell complexes from RD- and HFD-fed mice, with vehicle or antibiotic treatment at p7.

I   Quantification of MPs at p7 fold RD; n = 6 (RD), 4 (HFD), 3 (HFD+AB), 4 (RD+AB); CI 95%.

J, K   Representative FACS plots (J) and quantification of microglia (K) (Ly6G$^-$, F4/80$^+$, CD11b$^+$, CX3CR1$^{hi}$ CD45$^{lo}$) fold RD; n = 5 (RD), 5 (HFD), 3 (HFD+AB), 4 (RD+AB); CI 95%.

L   Representative confocal images of IBA-1-stained MPs on choroidal flat mounts from RD- and HFD-fed mice, treated with vehicle or with neomycin. Examples of labeled macrophages (white dots) are presented in side panels.

M   Total number of MPs around laser impact area; n = 7 (RD), 9 (HFD), 4 (HFD+AB), 7 (RD+AB), with 23 (RD), 29 (HFD), 13 (HFD+AB), 23 (RD+AB) burns total; CI 99%.

Data information: All comparisons between groups are analyzed using one-way analysis of variance (ANOVA) and Tukey's multiple comparisons test; *$P < 0.05$, **$P < 0.01$, ***$P < 0.001$; error bars represent mean ± SEM. Each "n" represents one mouse per experimental group; CI, confidence interval. Scale bar: 100 μm.

---

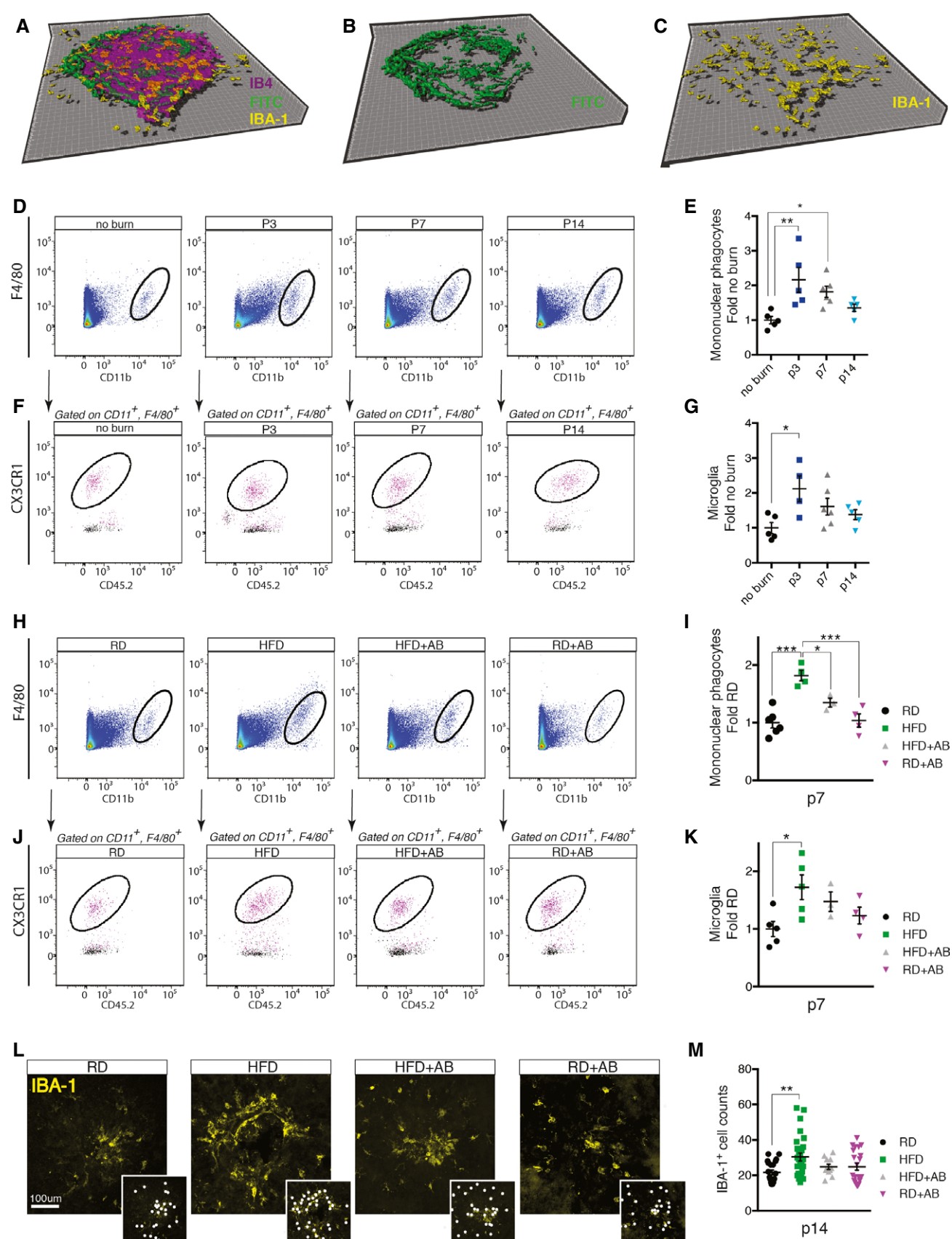

Figure 2.

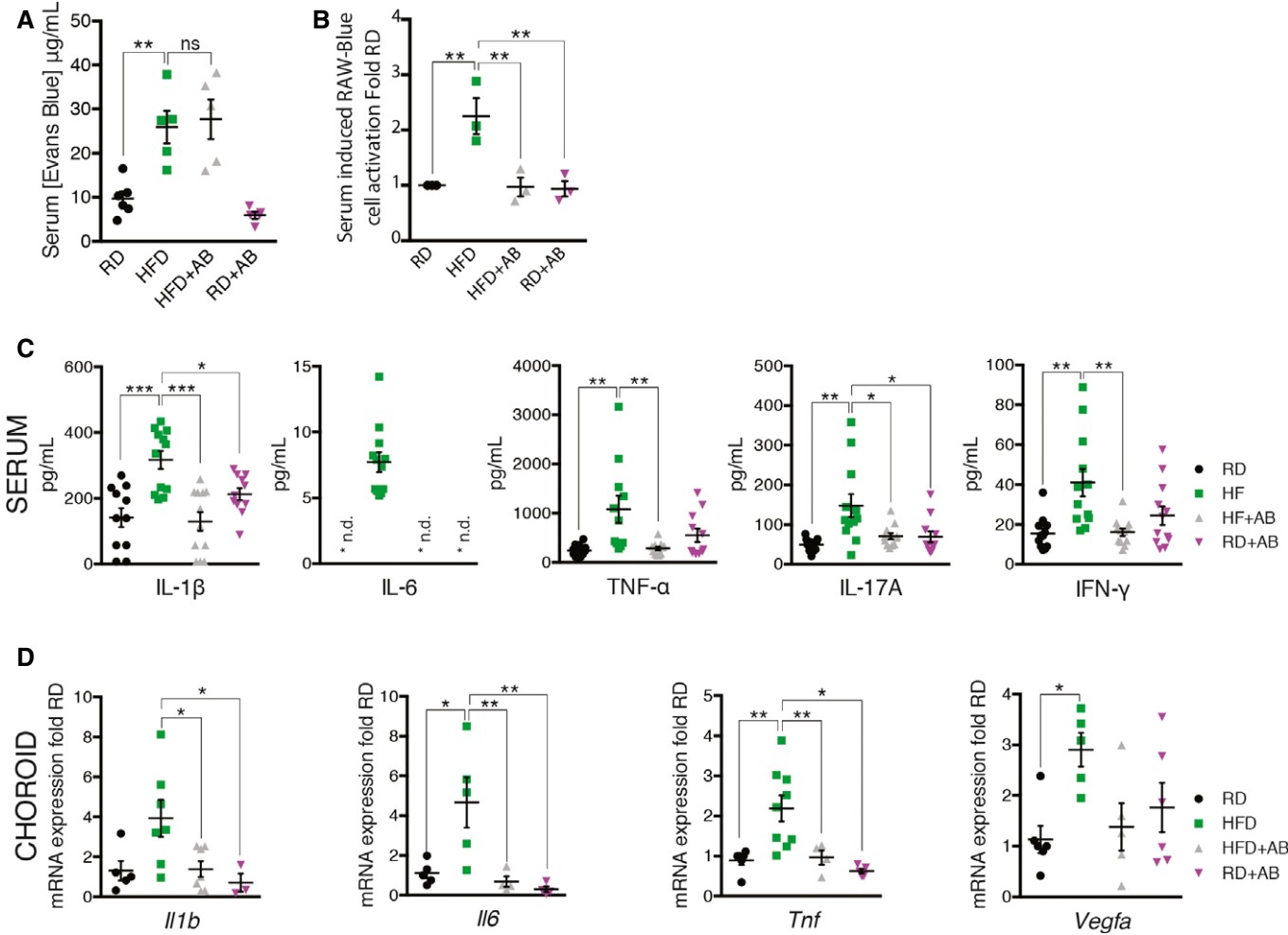

**Figure 3. High-fat diet augments gut permeability, metabolic endotoxemia, and systemic inflammation.**

A   Concentration of gut-absorbed Evans Blue in serum 24 h after oral administration in regular-chow diet (RD)- and high-fat diet (HFD)-fed mice, treated with vehicle or with antibiotic; $n = 6$ (RD), 5 (HFD), 5 (HFD+AB), 5 (RD+AB); CI 99%.

B   Activation of PRRs induced by stimulation with serum isolated from mice fed RD or HFD for 7 weeks and receiving vehicle or neomycin for 3 weeks; fold RD; $n = 3$ for all groups; CI 99%.

C   Serum cytokine profiles determined by Bio-Plex assay (Bio-Rad) of IL-1β, $n = 11$ (RD), 12 (HFD), 12 (HFD+AB), 11 (RD+AB); CI 95%; IL-6, $n = 11$ (RD), 12 (HFD), 12 (HFD+AB), 11 (RD+AB), CI 99.9%; TNF-α, $n = 11$ for all groups, CI 99%; IL-17A, $n = 11$ (RD), 12 (HFD), 12 (HFD+AB), 11 (RD+AB), CI 95%; and IFN-γ, $n = 12$ for all groups, CI 95%. *n.d. = not detected.

D   mRNA expression of *Il1b*, $n = 5$ (RD), 7 (HFD), 7 (HFD+AB), 3 (RD+AB), CI 95%; *Il6*, $n = 5$ (RD), 5 (HFD), 4 (HFD+AB), 4 (RD+AB), CI 95%; *Tnfa*, $n = 6$ (RD), 9 (HFD), 4 (HFD+AB), 5 (RD+AB), CI 95%; and *Vegfa*, $n = 6$ (RD), 5 (HFD), 5 (HFD+AB), 6 (RD+AB), CI 95% in choroids.

Data information: All comparisons between groups are analyzed using one-way analysis of variance (ANOVA) and Tukey's multiple comparisons test; *$P < 0.05$, **$P < 0.01$, ***$P < 0.001$; error bars represent mean ± SEM. Each "$n$" represents one mouse; CI: confidence interval.

activation of PRRs (Fig 3B), serum cytokine concentrations of IL-1β, IL-6, TNF-α, IL-17A, and IFN-γ were significantly induced with HFD, as was anti-inflammatory IL-10 (Figs 3C and EV3A). Oral intake of neomycin reversed the trend (Figs 3C and EV3A). Moreover, this pattern held in naive choroids (without laser-induced CNV) with transcript levels of *Il1b*, *Il6,* and *Tnfa* as well as with the cardinal angiogenic factor *Vegfa* (Fig 3D). Moreover, neomycin treatment of HFD-fed mice improved their glucose tolerance (Fig EV3B and C) suggesting a general improvement in metabolic health, an important risk factor for AMD (Ghaem Maralani *et al*, 2015), and confirming efficient modulation of gut microbes (Cani *et al*, 2007). These data suggest that HFD primes choroids for CNV in a gut flora-dependent

manner, where dysbiosis of the gut microbial community results in alterations of circulating levels of pro- and anti-inflammatory cytokines and the expression of inflammation-associated mRNAs.

## Microbiotal transplants confirm that high-fat diet aggravates CNV through gut microbiota

To ascertain that HFD-associated gut microbial communities drive increased CNV, we transplanted by oral gavage cecal microbiota harvested from the feces of RD-fed mice to HFD-fed mice (HFDxRDT). RD-fed mice transplanted with a fecal suspension of RD microbiota (RDxRDT) and HFD-fed mice transplanted with fecal

**Figure 4.  High-fat diet exacerbates CNV through gut microbiota.**

A    Schematic representation of microbiotal transfer experiments where recipient mice are gavaged with a suspension of fecal pellets in PBS from donor mice that are fed either a regular-chow diet (RD) or a high-fat diet (HFD).

B    Experimental timeline describing preparation of mice for microbiotal transfers where mice receive 5 days of antibiotics (neomycin and ampicillin) at 6 weeks of age. Starting at 7 weeks of life, RDxRDT and HFDxRDT mice receive weekly microbiotal transplants from RD donor mice and HFDxHFDT mice receive weekly microbiotal transplants from HFD donor mice until killing at week 13. At the age of 11 weeks, mice are subjected to four laser burns per eye.

C, D    Weight gain (C) and area under the curve (D) of percentage weight gain of RDxRDT, HFDxRDT, and HFDxHFDT mice; $n = 13$ (RDxRDT), 12 (HFDxHFDT), 14 (HFDxRDT); CI 99%.

E    Representative circle charts of relative abundance of bacterial phyla in gut microbiota of RD-fed mice receiving a RD transfer and HFD-fed mice receiving a HFD or RD transfer.

F    Relative proportion per experimental group of different phyla; $n = 5$ (RDxRDT), 5 (HFDxHFDT), 4 (HFDxRDT); Bacteroidetes; CI 99%, Firmicutes; CI 95%, Proteobacteria; CI 95%, Actinobacteria and Spirochaetes.

G    Compilation of compressed Z-stack confocal images of FITC–dextran-labeled CNV and single-plane confocal images of isolectin B4-stained choroidal flat mounts from RDxRDT, HFDxRDT, and HFDxHFDT mice.

H    Quantification of area of FITC–dextran-labeled CNV over area of isolectin B4-stained laser impact area fold RDxRDT; $n = 6$ (RDxRDT), 9 (HFDxHFDT), 9 (HFDxRDT), with 15 (RDxRDT), 21 (HFDxHFDT), 16 (HFDxRDT) burns total; CI 95%.

I    Concentration of gut-absorbed Evans Blue in serum 24 h after oral administration in RD- and HFD-fed mice, after RD or HFD microbiotal transplant; $n = 3$ (RDxHFDT), 3 (HFDxHFDT), 4 (HFDxRDT); CI 95%.

J    mRNA expression of *Il1b*: $n = 5$ (RDxRDT), 4 (HFDxHFDT), 5 (HFDxRDT); *Il6*: $n = 5$ (RDxRDT), 4 (HFDxHFDT), 4 (HFDxRDT), CI 99%; *Tnfa*: $n = 5$ (RDxRDT), 4 (HFDxHFDT), 4 (HFDxRDT), CI 95%; and *Vegfa*: $n = 5$ (RDxRDT), 4 (HFDxHFDT), 5 (HFDxRDT), CI 95% in choroids.

Data information: All comparisons between groups are analyzed using one-way analysis of variance (ANOVA) and Tukey's multiple comparisons test; *$P < 0.05$, **$P < 0.01$, ***$P < 0.001$; error bars represent mean ± SEM. Each "$n$" represents one mouse per experimental group; in (I), both eyes were used for mRNA extraction, giving two data points per mouse; CI, confidence interval. Scale bar, 100 μm.

suspension of HFD microbiota (HFDxHFDT) served as controls (Fig 4A). Prior to microbiotal transplantation, recipient mice were treated with ampicillin and neomycin in drinking water to deplete their original commensal microbiome. Gavage with cecal microbiota was repeated on a weekly basis to maintain a constant composition of the transplanted flora (Fig 4B). Although mice on HFD were significantly heavier than those on RD, microbiotal transfer from RD-fed mice did not significantly alter weight gain of HFD-fed mice and thus permitted to uncouple the effects of weight gain from those of microbial transfer (Fig 4C and D).

To verify successful transplantation of diet-associated microbial phyla, we characterized fecal microbiota as above (Figs 4E and EV4A). In RDxRDT mice, Bacteroidetes were the most abundant phylum representing 75–80% of the microbiome while accounting for only 35% in HFDxHFDT mice where Firmicutes and Proteobacteria filled the balance. Transfer of RD microbiota to HFD-fed mice restored microbial proportions found in RD-fed mice (Fig 4F). In addition, improved glucose tolerance in HFDxRDT mice compared to HFDxHFDT mice attests to successful microbiotal transplants (Fig EV4B and C).

While HFDxHFDT showed a ~twofold increase in CNV when compared to RDxRDT, transfer of RD microbiota to HFD-fed mice diminished CNV by ~35% supporting a role for gut microbiota in pathological CNV (Fig 4G and H). In addition, our data attest to intestinal permeability being lowered through microbiotal transfer. Whereas HFDxHFDT mice show twofold higher levels of serum Evans Blue compared to RDxRDT, transfer of RD microbiota to HFD-fed mice reduced permeability to levels seen in RDxRDT mice (Fig 4I). Analysis of choroidal levels of *Il6*, *Tnfa,* and *Vegfa* confirmed that transfer of RD-fed fecal suspensions to HFD mice reduced overall inflammation (Fig 4J). *Il1b* transcripts were not affected. In concert, these results provide evidence for the influence of gut microbiota on CNV.

## Discussion

With an increasing prevalence of obesity and increasing life expectancy, the societal impact and financial burden of AMD are expected to rise dramatically in the coming years (Friedman *et al*, 2004; Rein *et al*, 2006). Polymorphisms in genes implicated in inflammation predispose to AMD (Hageman *et al*, 2005) with strong linkage to single nucleotide polymorphisms in CFH factors, CFHR (CFH-related factors), ARMS2, VEGFA, and TGFBR1 (Ambati *et al*, 2003a; Klein *et al*, 2005; Yu *et al*, 2011; Ratnapriya & Chew, 2013), yet by themselves, no single mutation can account for disease development. To date, the contribution of our "second genome" (the microbiome) has not been investigated. Our study suggests that gut microbiota influences development of neovascular lesions associated with AMD and this is particularly when obesity is a predisposing factor. We show that diets rich in fat alter the gut microbiome and in turn elevate choroidal and systemic inflammation and heighten pathological choroidal neovascularization. This effect could originate from increased intestinal permeability to PAMPs secondary to dysbiosis (Fig 5). In this regard, obesity-related changes in gut microbiota have been shown to decrease barrier function of the gut, allowing increased entry of PAMPs into systemic circulation and consequently triggering inflammation (Amar *et al*, 2008; Cani *et al*, 2008; Osborn & Olefsky, 2012). Both antibiotic treatment and microbiotal transplants from RD-fed mice lower systemic and choroidal inflammation, yet only RD-microbiotal transplants significantly lower intestinal permeability. While the microbial community of HFDxRDT mice resembles the one of RD-fed mice more closely than that of HFD-fed mice treated with antibiotics, the effects of neomycin treatment likely occur through a decrease in the absolute number of bacteria present in the gut.

Modifying microbiota can reduce systemic and local choroidal inflammation and attenuate pathological neovascularization. Of note, one of the most heavily regulated inflammatory cytokines in our experimental paradigms was IL-6. Elevated levels of IL-6 are associated with early (Klein *et al*, 2014) and late AMD (Klein *et al*, 2008) and are significantly related to smoking, higher body mass index, and "inflammaging", the low-grade, chronic, systemic sterile inflammation in aging, in the absence of infection (Franceschi & Campisi, 2014). Dysbiosis of gut flora may thus be an additional factor that accounts for the inconsistent responses between

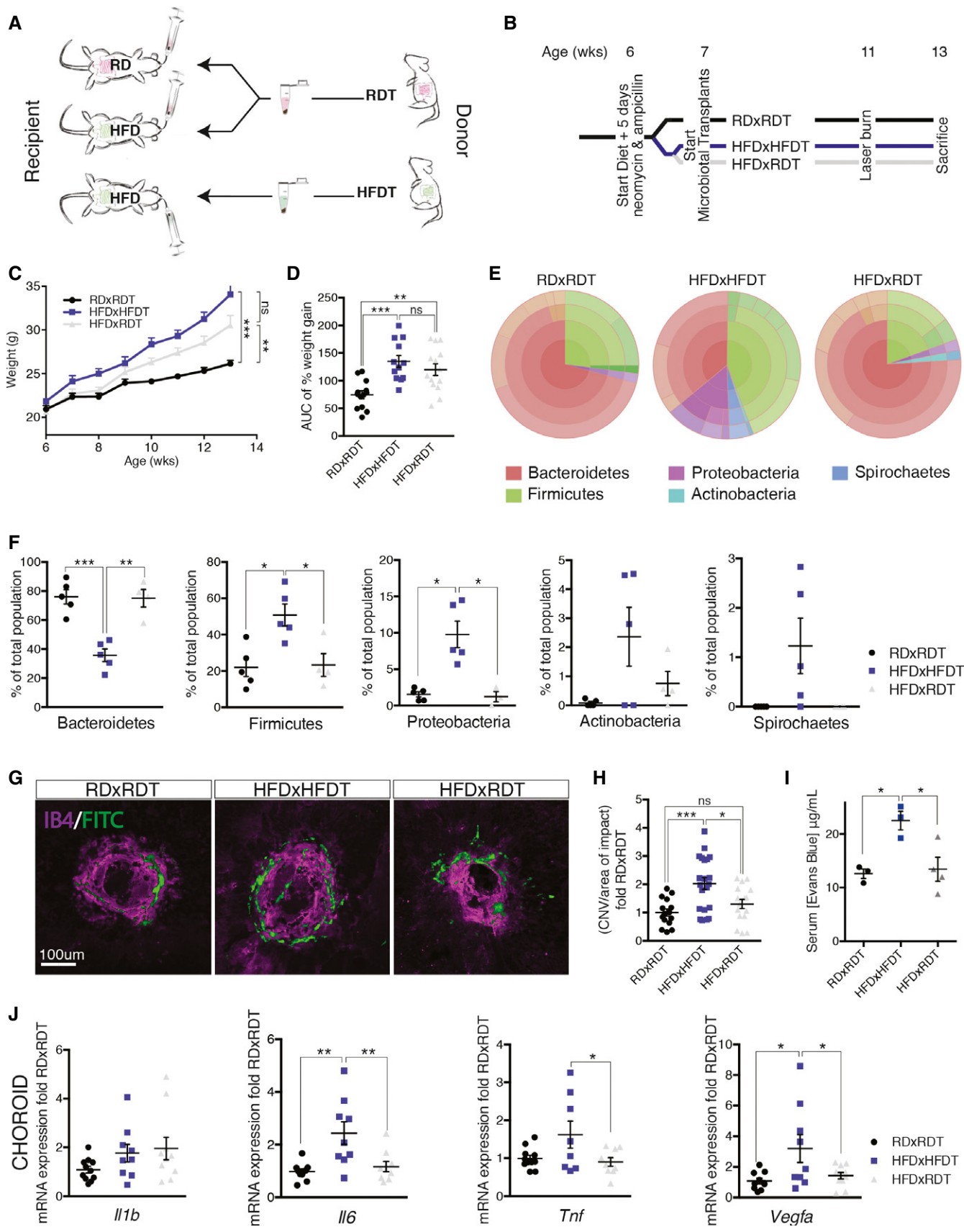

Figure 4.

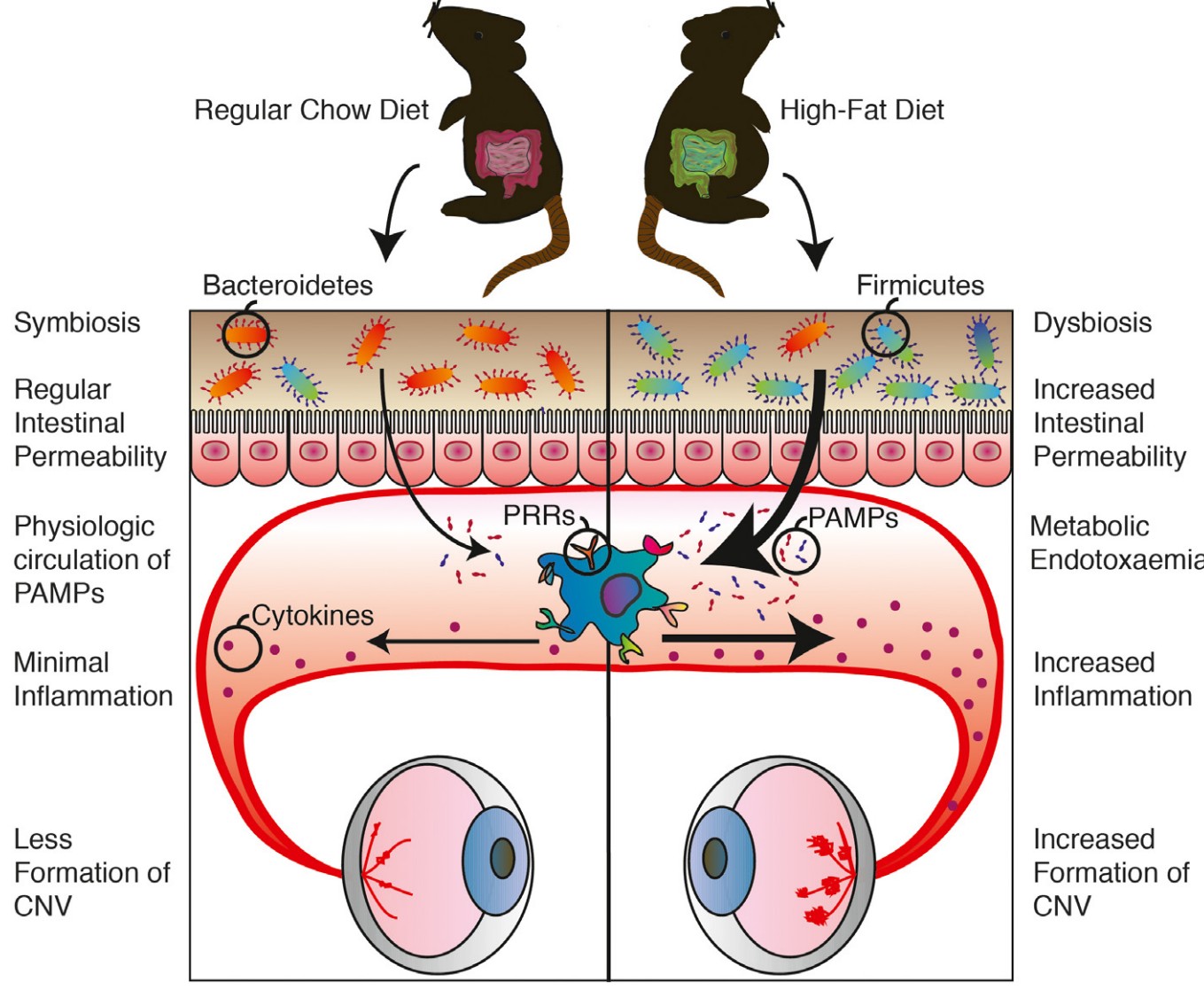

**Figure 5.  High-fat diet-induced dysbiosis increases intestinal permeability, metabolic endotoxemia, and systemic and local inflammation and ultimately contributes to CNV.**
The ratio of Bacteroidetes and Firmicutes, the two dominant phyla in intestinal flora, decreases significantly in high-fat diet (HFD)-fed mice compared to regular-chow diet (RD)-fed mice, with a relative increase in Firmicutes in HFD-fed mice and decrease in Bacteroidetes. This dysbiosis is accompanied by heightened intestinal permeability, which allows increased translocation of pathogen-associated molecular patterns (PAMPs) (endotoxemia). Recognition of these microbe-specific molecules by pattern recognition receptors (PRRs) present on innate immune cells triggers synthesis and excretion of cytokines causing chronic systemic low-grade inflammation. Ultimately, this exacerbates CNV.

individuals subjected to dietary interventions designed to stall progression of AMD such as AREDS formulations (Age-Related Eye Disease Study Research Group 2001; Age-Related Eye Disease Study 2 Research Group 2013). In addition, from an experimental perspective, our study strengthens the notion that housing and dietary considerations must be taken into account when designing animal-based studies on angiogenesis and aging. Gut microbiota is acquired shortly after birth from the surrounding environment, forming a relatively stable community that can shift composition under influence of environmental factors such as diet, exercise, and medication (Turnbaugh *et al*, 2009; David *et al*, 2014; Evans *et al*, 2014). Modifying gut microbiomes may thus provide minimally intrusive and cost-effective paradigms to prevent or delay exudative AMD.

## Materials and Methods

### Animals

All animal procedures were validated by the Animal Care Committee of the University of Montreal and Hôpital Maisonneuve-Rosemont in accordance with the guidelines established by the Canadian Council on Animal Care.

Briefly, 6-week-old male C57BL/6 WT mice, purchased from The Jackson Laboratory, raised under sterile barrier conditions and housed under a 12-h light cycle with water and food *ad libitum* were placed on either a regular-chow diet (RD) (16% kcal fat, 63% kcal carbohydrate, 21% kcal protein) or a high-fat diet (HFD) (60% kcal

fat, 26% kcal carbohydrate, 14% kcal protein) for 7 weeks and weighed weekly to register weight gain. At the age of 9 weeks, half of the RD and half of the HFD mice received antibiotic treatment with 0.5 g/l neomycin trisulfate salt hydrate in their drinking water, resulting in four different experimental groups: Control (RD), Antibiotics (RD+AB), High-Fat (HFD), and High-Fat + Antibiotics (HFD+AB). Neomycin is not absorbed from the gut (Cani *et al*, 2008); therefore, the antibiotic treatment was isolated to the GI tract and had no direct systemic effect.

At the age of 13 weeks, fecal pellets and blood were collected before sedation with isoflurane gas and cervical dislocation. Eyes were enucleated, dissected, and stored at −80°C for mRNA extraction.

### Laser-induced CNV

At the age of 11 weeks, mice were anesthetized with 10 μl/g body weight of a 10% ketamine and 4% xylazine solution, and their Bruch's membrane was ruptured using an argon laser as described previously (Lambert *et al*, 2013). Two weeks after CNV induction, the mice were perfused with 0.5 ml of 15 mg/ml of fluorescein isothiocyanate (FITC)–dextran (average mol wt 20,000) and killed. Eyes were enucleated and processed for analysis by immunohisto-chemistry. The sclera–choroid–RPE cell complex was mounted onto a slide, and the burns and macrophages were photographed with an Olympus FV1000 microscope. The neovascularization was captured in a Z-stack, and the lesion caused by the laser impact was captured in a single-plane image. The Z-stacks were compressed into one image, and the FITC–dextran-labeled neovascular area and the area of the lesion were measured per lesion in ImageJ.

### Immunohistochemistry

Eyes were fixed for 30 min in 4% PFA at room temperature, before dissection of the sclera–choroid–RPE cell complex. After a secondary fixation of 15 min in 4% PFA at room temperature, the choroids were stained with rhodamine-labeled Griffonia (Ban-deiraea) Simplicifolia Lectin I (Vector Laboratories Inc.) in 1 mM CaCl$_2$ in PBS and IBA-1 (rabbit polyclonal; Wako).

### Microbiome sequencing

DNA was extracted from fecal pellets with the Qiagen QIAamp Fast Stool Mini Kit according to the manufacturer's instructions with several small modifications; briefly, 2–3 fecal pellets were homogenized in 500 μl InhibitEX Buffer using a disposable homogenizing pestle and vortex. The suspension was heated for 5 min at 70°C before stool particles were pelleted by 1-min centrifugation at 20,000 *g*. The supernatant was thoroughly mixed with 20 μl proteinase K before 500 μl Buffer AL was added and the mix was incubated at 70°C for 10 min. After the addition of 500 μl of 100% ethanol, the lysate was applied to the QIAamp spin column and centrifuged for 1 min at 20,000 *g*. The filtrate was discarded before 500 μl Buffer AW1 was added to the spin column and centrifuged for 1 min at 20,000 *g*. This step was repeated with 500 μl Buffer AW2. The spin column was dried by centrifugation for 3 min at 20,000 *g* in a clean 2-ml collection tube. The DNA was eluted in 200 μl of Buffer ATE, directly pipetted on the QIAamp membrane, and collected in a clean Eppendorf tube.

After DNA extraction from the fecal pellets, the Thermo Fisher Ion 16S™ Metagenomics Kit was used to amplify the hypervariable regions V2, V3, V4, V6, V7, V8, and V9 of bacterial 16S rRNA. The amplified fragments were then bar-coded, sequenced on the Ion PGM sequencer system, and analyzed with the Ion Reporter Software.

### FACS

Retinas and sclera–choroid–RPE cell complexes of non-burned and burned mice at p3, p7, and p14 were cut into small pieces and homogenized in a solution of 750 U/ml DNase I (Sigma-Aldrich Corp., St. Louis, MO, USA) and 0.5 mg/ml of collagenase D (Roche, Basel, Switzerland) for 20 min at 37°C. Homogenates were filtered through a 70-μm cell strainer and washed in PBS. Viability of the cells was checked by Zombie Aqua (423101: Biolegend) staining for 15 min at room temperature. After incubation with LEAF-purified anti-mouse CD16/32 (101310; BioLegend, San Diego, CA, USA) for 15 min at room temperature to block Fc receptors, cells were incubated for 25 min at 4°C with the following antibodies: Alexa Fluor 700 anti-mouse CD45.2 (109821; BioLegend), BV711 anti-mouse/human CD11b (101241; BioLegend), APC/CY7 anti-mouse Ly-6G (127624; BioLegend), PE/CY7 anti-mouse F4/80 (123114; BioLegend), anti-mouse CX3CR1 phycoerythrin-conjugated goat IgG (FAB5825P; R&D Systems, Inc., Minneapolis, MN, USA). Fluorescence-activated cell sorting (FACS) was performed on a BD LSRFortessa™ X-20 cell analyzer, and data were analyzed using FlowJo software (version 7.6.5; FlowJo, Ashland, OR, USA).

### Intestinal permeability assay

Mice were injected by oral gavage with 1 ml of 50 mg/ml Evans Blue divided over five injections, 30 min apart. After 24 h, 120 μl of blood was collected from the submandibular vein. Serum was analyzed for Evans Blue concentration with a spectrophotometer at an optical density of 620–740 nm and quantified with the help of a standard dilution curve.

### RAW-blue assay

Blood was collected through cardiac puncture, and serum was stored at −80°C until use in the RAW-Blue assay. PAMPs were assayed with RAW-Blue™ cells (InvivoGen, San Diego, CA) through detection of NF-κB/AP-1 activation following activation of TLRs (with the exception of TLR5), NOD1/2, and dectin-1 using a modified version of the manufacturer's protocol. RAW-Blue™ cells were grown in growth medium (DMEM, 4.5 g/l glucose, 2 mM L-glutamine, 10% fetal bovine serum (FBS), 100 μg/ml Zeocin™ (InvivoGen, San Diego, CA)). The assay was performed when the cells were in passage 10–15 by plating $10^5$ cells in 96-well plates containing basal DMEM. After 6 h of starvation 30 μl of mouse serum, 30 μl of FBS(−control) or 30 μl of FBS+LPS (+ control) was added per well and cells were incubated for 21 h at 37°C under an atmosphere of 5% CO$_2$/95% air. SEAP levels were determined using a spectrophotometer at 620–655 nm after a 1- to 3-h incubation at 37°C of 20 μl of induced RAW-Blue™ cell supernatant with 180 μl QUANTI-Blue™ (InvivoGen, San Diego, CA).

## Cytokine assessment

Serum cytokine assays were performed using a Bio-plex Mouse Cytokine 6-plex panel (1 × 96-well) (Bio-Rad) according to the manufacturer's instructions. The Bio-Plex cytokine assay is a multiplex bead-based assay involving matrices that is designed to quantitate multiple cytokines as follows. The wells of a 96-well plate were pre-wet with 100 μl of Bio-Plex assay buffer. 50 μl of vortexed multiplex bead working solution was pipetted into each well and immediately removed. Wells were washed twice with the Bio-Plex wash buffer before 25 μl of vortexed Bio-Plex Detention Antibody working solution was added to each well and incubated for 30 min. After a triple wash with the Bio-Plex wash buffer, 50 μl of vortexed 1× streptavidin-PE was added to each well and incubated for 30 min. After three washes, the beads in each well were resuspended with 125 μl of Bio-Plex assay buffer and vortexed for 30 s, and the plate was immediately read on the Bio-Plex system. Cytokine concentrations were calculated from the standard curve by the use of Bio-Plex manager software. Samples were run in duplicate.

## Real-time PCR analysis

After enucleation eyes were dissected to isolate retinas and sclera–choroid–RPE cell complex. RNA was isolated using TRIzol and digested with DNase I to prevent amplification of genomic DNA contaminants. M-MLV reverse transcriptase (Life Technologies) was used for the reversed transcription, and SYBR Green (Bio-Rad) to determine gene expression in an ABI Biosystems Real-Time PCR machine with β-actin (*Actb*) as a reference gene. We used the following primers: Mouse *Actb* = F: 5′-GAC GGC CAG GTC ATC ACT ATT G-3′, R: 5′-CCA CAG GAT TCC ATA CCC AAG A-3′; Mouse *Il1b* = F: 5′-CTG GTA CAT CAG CAC CTC ACA-3′, R: 5′-GAG CTC CTT AAC ATG CCC TG-3′; Mouse *Il6* = F: 5′-AGA CAA AGC CAG AGT CCT TCA GAG A-3′, R: 5′-GCC ACT CCT TCT GTG ACT CCA GC-3′; Mouse *Tnfa* = F: 5′-CCC TCA CAC TCA GAT CAT CTT CT-3′, R: 5′-GCT ACG ACG TGG GCT ACA G-3′; and Mouse *Vegfa* = F: 5′-GCC CTG AGT CAA GAG GAC AG-3′, R: 5′-CTC CTA GGC CCC TCA GAA GT-3′.

## Microbiota transplant

A litter of six C57BL/6 WT mice was divided over three cages, representing three experimental groups: regular-chow diet-fed mice receiving a transplant from regular-chow diet-fed mice (RDxRDT), high-fat diet-fed mice receiving a transplant from high-fat diet-fed mice (HFDxHFDT), and high-fat-fed mice receiving a transplant from regular-chow diet-fed mice (HFDxRDT). At the age of 6 weeks, two cages were started on HFD, while the other cage remained on a regular-chow diet. Mice received antibiotics (ampicillin 1.0 g/l + neomycin 0.5 g/l) in their drinking water for 5 days, to deplete the original gut microbiota, as described before (Ellekilde *et al*, 2014). Two days after discontinuing the antibiotics, they received their first microbiota transplant, from mice on either a regular-chow diet or a high-fat diet.

Feces were collected from 20 different donor mice, either on RD or on HFD for at least 5 weeks. The fecal pallets were collected in 2.5 ml of sterile PBS with 0.05% cysteine HCL, homogenized, and centrifuged at 200 *g* for 5 min. 200 μl of the supernatant was

**The paper explained**

**Problem**

Age-related macular degeneration in its neovascular form (NV AMD) is the leading cause of vision loss among adults above the age of 60. Epidemiological data suggest that in men, overall abdominal obesity is the second most important environmental risk factor after smoking for progression to late-stage NV AMD. To date, the mechanisms that underscore this observation remain ill-defined. Given the impact of high-fat diets on gut microbiota, we investigated whether commensal microbes influence the evolution of AMD.

**Results**

Using mouse models of NV AMD, microbiotal transplants and other paradigms that modify the gut microbiome, we uncoupled weight gain from confounding factors and demonstrate that high-fat diets exacerbate choroidal neovascularization (CNV) by altering gut microbiota. Gut dysbiosis leads to heightened intestinal permeability and chronic low-grade inflammation with elevated production of IL-6, IL-1β, TNF-α and VEGF-A that ultimately exacerbate pathological angiogenesis.

**Impact**

With an increasing prevalence of obesity and increasing life expectancy, the societal impact and financial burden of AMD are expected to rise dramatically in the coming years. Our study suggests that gut microbiota influences development of neovascular lesions associated with AMD, particularly when obesity is a predisposing factor. Modifying microbiota can reduce systemic and local choroidal inflammation and attenuate pathological neovascularization and may thus provide minimally intrusive and cost-effective paradigms to prevent or delay exudative AMD.

introduced orally to the mice with the help of flexible gavage needles. This was repeated with fresh fecal samples every 7 days for the duration of the experiment. At the age of 11 weeks, mice were subjected to four laser burns per eye. Mice were weighed weekly and feces were collected for analysis by 16S sequencing. Mice are coprophagic, and hence, care is taken to isolate individuals with given gut microbiota modifying protocols.

## Glucose tolerance test (GTT)

Mice were starved for 12 h overnight. Blood glucose was measured (Accu-Chek; Roche) at baseline and 15, 30, 60, 120, and 240 min following intraperitoneal injection of 2 mg/kg of 10% D-glucose.

## Statistical analysis

Results are presented as mean ± SEM. GraphPad Prism version 6.00 (GraphPad Software, San Diego, CA; www.graphpad.com) was used to analyze the statistical significance of differences by one-way ANOVA and Tukey's multiple comparisons test or a two-tailed Student's *t*-test, where appropriate. Data with $P < 0.05$ were considered statistically different: *$P < 0.05$, **$P < 0.01$, and ***$P < 0.001$.

**Expanded View** for this article is available online.

## Acknowledgements

E.A. holds scholarships from the Réseau en Recherche en Santé de la Vision and the Faculté des Études Supérieures de l'UdM. P.S. holds The

Wolfe Professorship in Translational Vision Research and the Canada Research Chair in Retinal Cell Biology and the Alcon Research Institute Young Investigator Award. This work was supported by grants to P.S. from The Foundation Fighting Blindness Canada (FFB), the Canadian Institutes of Health Research (324573), the Canadian Diabetes Association (OG-3-11-3329-PS), and The Natural Sciences and Engineering Research Council of Canada (418637) and the Fondation de l'Avenir, Paris, France, Etude n°AP-RMA-2015-010 to FS.

## Author contributions

PS and EMMAA designed the study. EMMAA, AMW, GM, AD, and KM performed experiments. FS provided valuable insight and comments on study design and the manuscript. EMMAA and PS wrote the manuscript. All authors discussed the results and commented on the manuscript at all stages.

## Conflict of interest

The authors declare that they have no conflict of interest.

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
