## [Review Process File · EMBO Molecular Medicine]

Gut Microbiota Influences Pathological Angiogenesis in Obesity-driven Choroidal Neovascularization

Elisabeth Andriessen, Ariel Wilson, Miss Gaelle Mawambo, Agnieszka Dejda, Khalil Miloudi, Florian Sennlaub, and Przemyslaw Sapieha

Corresponding author: Przemyslaw Sapieha, University of Montreal

Review timeline:

Submission date:	28 April 2016
Editorial Decision:	06 July 2016
Revision received:	21 September 2016
Editorial Decision:	05 October 2016
Revision received:	06 October 2016
Accepted:	11 October 2016

Transaction Report:

Editor: Céline Carret

1st Editorial Decision

06 July 2016

Thank you for the submission of your manuscript to EMBO Molecular Medicine. We have now heard back from the three referees whom we asked to evaluate your manuscript. Although the referees find the study to be of potential interest, they also raise a number of concerns that need to be addressed in the next final version of your article.

You will see that the referees find the study to be of interest, but regret poor citing and discussing of previous published work and ref3 suggestions should be taken into account. Additional experiments would be needed to add more data on mechanisms and thereby increase the conclusiveness of the findings. Finally, we would like to suggest changing the format from "Report" to "Article" in order to better reflect the current knowledge in the literature.

Given the balance of these evaluations, we feel that we can consider a revision of your manuscript if you can address the issues that have been raised within the space and time constraints outlined below. Please note that it is EMBO Molecular Medicine policy to allow only a single round of revision and that, as acceptance or rejection of the manuscript will depend on another round of review, your responses should be as complete as possible. Revised manuscripts should be submitted within three months of a request for revision; they will otherwise be treated as new submissions, except under exceptional circumstances in which a short extension is obtained from the editor.

I look forward to seeing a revised form of your manuscript as soon as possible.

***** Reviewer's comments *****

Referee #1 (Remarks):

Andriessen et al. have analyzed the relation of obesity, gut microbiota and the development of laser-induced choroidal neovascularization, an established model mimicking features of age-related macular degeneration. The study is principally of high interest as a direct connection of all these three aspects has not been demonstrated before despite publications of several human epidemiological data linking overweight with late AMD progression. The paper is principally well written but lacks precision in wording of introduction and conclusion. Several major issues should be addressed to support the authors conclusions and to make the data more solid regarding the molecular and cellular mechanisms.

Points of criticism:

1. Title: The title states that obesity exacerbates AMD by altering gut microbiota. The wording must be used more carefully as the authors did not perform a study in AMD patients (as implicated by the title) but analyzed the laser-CNV model. This model is indeed a relatively good model for nvAMD but can only reflect some issues related to immunity and angiogenesis.
2. Wrong and missing references in the introduction/discussion. I was surprised when I noticed that the authors used several citations that did not reflect the meaning of the written statements properly. Ref #2 is a paper on diabetic retinopathy and not related to AMD, ref #13 does not list increased systemic cytokine levels in AMD, ref #17 is on xenobiotic metabolism and does not discuss inflammation. The following important references have not been included in the introduction or in the discussion: 1. the newest genetic paper on AMD risk genes is missing (Fritsche LG et al Nat Genet 2016), 2. Zhang et al. IOVS 2016 57:1276 and 3. Maralani et al. Retina 2015 35:459 provide comprehensive meta-analyses of large prospective studies relating obesity/metabolic syndrome with AMD and should be included. 4. Horai et al. Immunity 2015 reports on microbiota-dependent immune activation in the retina, this is directly related to the topic presented here. 5. An ARVO abstract from 2007 annual meeting (Vol 48, 1768) also found that high fat diet increased leaky lesions and CNV size in the laser model.
3. In most figure with mean value bar graphs: Individual data points (either mean values per eye or individual lesions) should be plotted. This is important as the reader can directly estimate n-numbers and variation in the data set.
4. How was CNV size determined in Figs 1D, 3E, per eye ? per lesion ? The materials/methods section is not detailed enough here.
5. Vascular leakage data are usually determined to complement flat mount stainings of IB4/FITC. Sometimes inflammation-driven leakage is not identical with neovessels.
6. Fig 2A/B would significantly profit from a more sophisticated analysis of mononuclear phagocyte reactivity (microglia vs macrophages) and if possible complete retinal sections to demonstrate the location of cells in the inner vs outer retina.
7. It is left open how the changed microbiota may contribute to the different aspects of disease progression in this model. Given the recent findings from Erny et al. Nat Neuroscience 18:965 and Horai et al. 2015 Immunity one could hypothesize that resident retinal microglia have been primed by the changed gut microbes before the laser CNV model was applied. In this sense, it would be good if the authors could expand their cytokine expression analysis in serum (ELISA) and choroid (RNA) to isolated (FACS) immune cells and immune marker expression profiling of retina vs choroid/RPE. An additional time point where inflammation is maximum (e.g. at days 3 or 7) would be better suited than day 14 where most immune events have already calmed down.

Referee #2 (Comments on Novelty/Model System):

While laser induced CNV is often questioned as a model for CNV in AMD it is certainly the most widely used one.

Referee #2 (Remarks):

The manuscript raises very interesting issues about multifactorial diseases like AMD. I found the paper straight forward and very well argues. I do not find any issues to be raised.

Referee #3 (Remarks):

The authors examine the role of microbiota in influencing CNV (not AMD) in a high fat diet model. There is a vast body of literature other than the body mass indices that implicates abnormalities in lipid homeostasis in AMD. Multiple GWAS have demonstrated polymorphisms in numerous genes involved in regulation of cholesterol homeostasis. The clear message from these studies is that the polymorphisms are suggestive of a complex interaction at a tissue level.

The authors demonstrate that there is increased CNV after a high fat diet in the laser CNV model and state that this is the first demonstration of this. There is a highly cited study in Cell Metabolism from 2013 that shows that in diet induced obesity there is increased CNV. This was a seminal paper in the field and as such these data are not novel. The authors should conduct a careful reading of the literature so that appropriate credit can be given to prior literature.

RD is a confusing term for regular chow as it is too similar to conditions of the retina such as retinal detachments and retinal degenerations that are also called RD.

A 35% reduction in CNV in a mouse laser induced CNV model with high variability is unlikely to be clinically relevant. The molecular mechanisms behind this effect are not elucidated in this study and given that the finding that mice on high fat diets are prone to increased CNV is not a novel finding, it would be interesting to see at a molecular level what the effects are on lipid composition and how it influences CNV.

1st Revision - authors' response

21 September 2016

Detailed response to reviewers

Reviewer #1:

We thank the reviewer for their thoughtful comments and positive assessment of the study and thank them for acknowledging that “this study is principally of high interest.” Based on the recommendations and queries of the reviewer, we have performed a series of new experiments. Most notably, in the revised manuscript, we now provide a new mechanism of action for diete-induced para-inflammation by demonstrating elevated gut permeability in mice receiving high fat diets. We demonstrate that this heightened permeability can be rescued with microbial transplants. We also now profile infiltration of mononuclear phagocytes and microglia in different dietary paradigms by FACS.

Query 1. *Title: The title states that obesity exacerbates AMD by altering gut microbiota. The wording must be used more carefully as the authors did not perform a study in AMD patients (as implicated by the title) but analyzed the laser-CNV model. This model is indeed a relatively good model for nvAMD but can only reflect some issues related to immunity and angiogenesis.*

Response- We thank the reviewer for bringing-up this important point. We have now modified the title to: Gut Microbiota Influences Pathological Angiogenesis in Obesity-driven Choroidal Neovascularization

Query 2. *Wrong and missing references in the introduction/discussion.*

Response – We are grateful to the reviewer for bringing these studies to our attention and we have incorporated all references suggested by the reviewer and more into our revised manuscript. We also modified the incorrect ones.

Query 3. *In most figure with mean value bar graphs: Individual data points (either mean values per eye or individual leasons) should be plotted. This is important as the reader can directly estimate n-numbers and variation in the data set.*

Response- We fully agree with the reviewer and have proceed to modify all graphs to scatter plots in order better visualize the spread of data and number of data points.

Query 4. *How was CNV size determined in Figs 1D, 3E, per eye ? per lesion ? The materials/methods section is not detailed enough here.*

Response – We agree and have added additional information on the methodology in the ‘Methods’ section of the paper: The neovascularization was captured in a Z-Stack, and the lesion caused by the laser impact was captured in a single plane image. The Z-stacks were compressed into one image and the FITC-dextran labeled neovascular area and the area of the lesion were measured per lesion in Image-J.

Query 5. *Fig 2A/B would significantly profit from a more sophisticated analysis of mononuclear phagocyte reactivity (microglia vs macrophages).*

Response- We agree with the reviewer and we conducted an extensive characterization of infiltrating immune cells by FACS in our dietary paradigms. The new data is presented in the revised Figure 2.

Query 6. *... it would be good if the authors could expand their cytokine expression analysis in serum (ELISA) and choroid (RNA) to isolated (FACS) immune cells and immune marker expression profiling of retina vs choroid/RPE. An additional time point where inflammation is maximum (e.g. at days 3 or 7) would be better suited than day14 where most immune events have already calmed down.*

Response- We thank the reviewer for this suggestion. We addressed both points in the revised manuscript. First, we profiled by FACS infiltration of MPs and Microglia at 3, 7 and 14 days as suggested (New Figure 2). We also profiled inflammatory genes in the choroids of mice with different dietary paradigms (New Figure 2) and after microbial transfer (New Figure 4).

Reviewer #2

We thank the reviewer for their very positive evaluation and appraisal of the study and for acknowledging that: “The manuscript raises very interesting issues about multifactorial diseases like AMD ». The reviewer « did not find any issues to be raised.»

Reviewer #3

We thank the reviewer for their insightful, very helpful and pertinent comments. We believe that the reviewer’s suggestion to further explore a mechanism was significantly beneficial for the current study. We were planning on keeping portions of this mechanism for a follow-up article but agree that it strengthens the current study. We now provide evidence that the gut microbial dysbiosis caused by high-fat diets heightens intestinal permeability leading to increased circulating pathogen associated molecular patterns PAMPs, and a inflammatory response through pattern recognition receptors.

Query 1. *There is a highly cited study in Cell Metabolism from 2013 that shows that in diet induced obesity there is increased CNV. This was a seminal paper in the field and as such these data are not novel. The authors should conduct a careful reading of the literature so that appropriate credit can be given to prior literature.*

Response – We agree with the reviewer and now cite the seminal and highly cited Cell Metabolism paper from the RJ Apte group. We also now reference additional recent studies that highlight obesity as a predisposing factor to late AMD such as Maralani et al. (Retina 2015 35:459) and Zhang et al. (IOVS 2016 57:1276).

Query 2. *RD is a confusing term for regular chow as it is too similar to conditions of the retina such as retinal detachments and retinal degenerations that are also called RD.*

Response – We thank the reviewer for bringing this point to our attention. We have now carefully included text explaining that RD is short for Regular Feed in all legends and part of the text.

Query 3. *A 35% reduction in CNV in a mouse laser induced CNV model with high variability is unlikely to be clinically relevant.*

Response – Thank you for the comment. If this study were testing a drug or treatment paradigm, then the reviewer is absolutely correct. However, obesity is condition that will affect para-inflammation in a very protracted manner. Hence even smaller effects may have compounded outcomes in a disease with long-term outcomes such as AMD.

Query 4. *The molecular mechanisms behind this effect are not elucidated in this study.*

Response: We agree with the reviewer. We now provide evidence that the gut microbial dysbiosis caused by high-fat diets results in heightened intestinal permeability which increases circulating pathogen associated molecular patterns (PAMPs), leading to low-grade endotoxemia that triggers an inflammatory response through pattern recognition receptors (PRRs). Ultimately, this exacerbates choroidal neovascularization. We provide evidence for this mechanism in figure 3 and a rescue mechanism with microbiotal transfer in figure 4.

2nd Editorial Decision

05 October 2016

Thank you for the submission of your revised manuscript to EMBO Molecular Medicine. We have now received the enclosed report from the referee who was asked to re-assess it. As you will see this reviewer is now fully supportive and I am pleased to inform you that we will be able to accept your manuscript pending final editorial amendments.

Please submit your revised manuscript within two weeks. I look forward to seeing a revised form of your manuscript as soon as possible.

***** Reviewer's comments *****

Referee #1 (Comments on Novelty/Model System):

The techniques including the laser CNV-model and the microbiota analyses are state of the art.

Referee #1 (Remarks):

The authors have taken the comments of the reviewers very seriously and addressed all points appropriately. I think that they did a very good job by performing additional experiments to further characterize the relevant immune cell populations by stainings and FACS assays. The flow of the paper is now also much better as key references have been included and the methods are given in greater detail. Overall, this is a highly interesting study that clearly shows the direct influence of gut microbiota on angiogenesis in the posterior eye.

YOU MUST COMPLETE ALL CELLS WITH A PINK BACKGROUND ↓
PLEASE NOTE THAT THIS CHECKLIST WILL BE PUBLISHED ALONGSIDE YOUR PAPER

Corresponding Author Name: Sapielha Przemyslaw
Journal Submitted to: EMBO Molecular Medicine
Manuscript Number: EMM-2016-06531